# Potential Oral Microbial Markers for Differential Diagnosis of Crohn’s Disease and Ulcerative Colitis Using Machine Learning Models

**DOI:** 10.3390/microorganisms11071665

**Published:** 2023-06-26

**Authors:** Sang-Bum Kang, Hyeonwoo Kim, Sangsoo Kim, Jiwon Kim, Soo-Kyung Park, Chil-Woo Lee, Kyeong Ok Kim, Geom-Seog Seo, Min Suk Kim, Jae Myung Cha, Ja Seol Koo, Dong-Il Park

**Affiliations:** 1Department of Internal Medicine, College of Medicine, Daejeon St. Mary’s Hospital, The Catholic University of Korea, Daejeon 34943, Republic of Korea; dxandtx@catholic.ac.kr; 2Department of Bioinformatics, Soongsil University, Seoul 06978, Republic of Korea; blissfulwooji@gmail.com (H.K.); sskimb@ssu.ac.kr (S.K.); kjwk1221@gmail.com (J.K.); 3Division of Gastroenterology, Department of Internal Medicine and Inflammatory Bowel Disease Center, Kangbuk Samsung Hospital, School of Medicine, Sungkyunkwan University, Seoul 03181, Republic of Korea; sk0103.park@samsung.com; 4Medical Research Institute, Kangbuk Samsung Hospital, School of Medicine, Sungkyunkwan University, Seoul 03181, Republic of Korea; chilwoo.lee@gmail.com; 5Department of Internal Medicine, College of Medicine, Yeungnam University, Daegu 42415, Republic of Korea; cello7727@naver.com; 6Department of Internal Medicine, School of Medicine, Wonkwang University, Iksan 54538, Republic of Korea; medsgs@wku.ac.kr; 7Department of Human Intelligence and Robot Engineering, Sangmyung University, Cheonan-si 31066, Republic of Korea; minsuk.kim@smu.ac.kr; 8Department of Internal Medicine, Kyung Hee University Hospital at Gangdong, Kyung Hee University College of Medicine, Seoul 05278, Republic of Korea; clicknox@khnmc.or.kr; 9Division of Gastroenterology and Hepatology, Department of Internal Medicine, Ansan Hospital, Korea University College of Medicine, Ansan 15355, Republic of Korea; jskoo@korea.ac.kr

**Keywords:** oral microbiome, dysbiosis, 16S rRNA sequencing, IBD, machine learning, sPLS-DA

## Abstract

Although gut microbiome dysbiosis has been associated with inflammatory bowel disease (IBD), the relationship between the oral microbiota and IBD remains poorly understood. This study aimed to identify unique microbiome patterns in saliva from IBD patients and explore potential oral microbial markers for differentiating Crohn’s disease (CD) and ulcerative colitis (UC). A prospective cohort study recruited IBD patients (UC: *n* = 175, CD: *n* = 127) and healthy controls (HC: *n* = 100) to analyze their oral microbiota using 16S rRNA gene sequencing. Machine learning models (sparse partial least squares discriminant analysis (sPLS-DA)) were trained with the sequencing data to classify CD and UC. Taxonomic classification resulted in 4041 phylotypes using Kraken2 and the SILVA reference database. After quality filtering, 398 samples (UC: *n* = 175, CD: *n* = 124, HC: *n* = 99) and 2711 phylotypes were included. Alpha diversity analysis revealed significantly reduced richness in the microbiome of IBD patients compared to healthy controls. The sPLS-DA model achieved high accuracy (mean accuracy: 0.908, and AUC: 0.966) in distinguishing IBD vs. HC, as well as good accuracy (0.846) and AUC (0.923) in differentiating CD vs. UC. These findings highlight distinct oral microbiome patterns in IBD and provide insights into potential diagnostic markers.

## 1. Introduction

Because IBD can cause irreversible damage to the digestive tract if left untreated, early diagnosis and treatment should be mandatory to control inflammation and potentially limit this damage [1,2,3]. To diagnose IBD in the early stages, it is crucial for physicians to conduct a careful and thorough evaluation, which includes history-taking, laboratory tests, endoscopy, and cross-sectional imaging [4,5,6].

IBD is a complex disorder that involves the interplay of several factors: genetics [7], the immune system, environmental factors, and the gut microbiome. Among these factors, dysbiosis, which refers to the imbalance of microbial communities, is believed to play a significant role in the pathogenesis of IBD [8,9]. However, the relationship between dysbiosis and IBD is currently complex and not fully understood. While there is a clear association between dysbiosis and IBD, it remains uncertain whether dysbiosis is a cause or a consequence of IBD [10]. Dysbiosis can occur not only in the gut, but also in the oral cavity, potentially influencing the function of the gut barrier and immune response. These factors play a crucial role in the development of chronic inflammation, which is a characteristic feature of IBD [11,12].

According to recent studies [13,14,15], exploring the oral microbiome as a potential diagnostic tool for IBD offers several advantages, including ease of sample collection, stability, and early detection compared to fecal microbiome analysis. These studies have revealed a connection between the oral microbiome and IBD, as individuals with IBD tend to harbor distinct bacterial profiles in their mouths compared to those without the condition [16].

However, it is important to acknowledge the limitations and challenges associated with analyzing the oral microbiome for IBD diagnosis. Saliva-based microbiome analysis is a complex process that can be influenced by numerous factors, making it difficult to identify clear, disease-specific patterns. Furthermore, the oral microbiome may not directly reflect the gut environment as accurately as the fecal microbiome, potentially providing less relevant information about the disease state. To overcome these challenges, microbial analysis utilizing algorithmic machine learning models can be beneficial. These models address issues such as data complexity, the requirement for large and high-quality datasets, and validation problems, as well as ethical and privacy considerations [17,18,19]. One statistical method commonly used in this context is sparse partial least squares discriminant analysis (sPLS-DA), which aims to establish a linear relationship between two datasets while performing variable selection. This approach can help identify the most important microbial species for predicting the disease outcome [20].

The aim of this study was to assess the microbial patterns in saliva and identify differences between healthy controls and patients with IBD. Additionally, the study aimed to utilize a machine learning model, specifically sPLS-DA, to identify subgroups within IBD, such as CD and UC.

## 2. Materials and Methods

### 2.1. Study Population and Sample Collection

We recruited two cohorts of patients—one with CD and the other with UC—as well as a cohort of healthy controls for our study. The patients with CD were included from 15 tertiary hospitals in South Korea as part of a multicenter, retrospective case–control study [21]. For the ulcerative colitis (UC) multiomics study, a prospective multicenter study was established in Korea in 2020. A total of 14 university hospitals participated in this study and collected clinical data and biological specimens, including blood, stool, tissue, and saliva samples, from UC patients. From the outpatients who underwent screening by colonoscopy between June 2020 and December 2020, we selected 100 individuals who agreed to participate in the study and provide saliva samples to form the healthy control group.

Saliva samples (2 mL) were collected using a saliva collection kit (Cat. PDX-026; PDXen Biosystems Co., Daejeon, Republic of Korea) after lightly rinsing the mouth twice with mineral water, and they were transported and stored at room temperature (15–30 °C).

### 2.2. DNA Extraction, PCR Amplification, and 16S rRNA Gene Sequencing

Details of the sample preparation and sequencing have been published in [22]. In summary, the samples underwent centrifugation at 15,000 rpm for 20 min at 4 °C to separate the cellular pellet from the cell-free supernatant. Subsequently, DNA extraction from the cellular pellet was carried out using the QIAamp DNA Microbiome Kit (Qiagen, Valencia, CA, USA), following the manufacturer’s protocol.

The V3-4 region of the 16S rRNA gene was amplified using the 341F (5′-TCG TCG GCA GCG TCA GAT GTG TAT AAG AGA CAG CCT ACG GGN GGC WGC AG-3′) and 805R (5′-GTC TCG TGG GCT CGG AGA TGT GTA TAA GAG ACA GGA CTA CHV GGG TAT CTA ATC C-3′) primers, which included Illumina adaptor overhang sequences. The amplicons generated through PCR were purified using a magnetic-bead-based clean-up system (Agen-court AMPure XP Beckman Coulter^TM^, Brea, CA, USA). Indexed libraries were prepared using Nextera technology, which involves limited-cycle PCR, and the libraries were subsequently cleaned up and pooled at equimolar concentrations. Paired-end sequencing was performed on the Illumina MiSeq platform using a 2 × 300 bp protocol, following the manufacturer’s instructions.

### 2.3. 16S rRNA Gene Sequencing Data Analysis

The raw sequence data underwent quality filtering using FASTP (version 0.23.2, https://github.com/OpenGene/fastp, accessed on 16 August 2022) to remove sequences with low quality (quality score < 20) and reads shorter than 150 base pairs. Taxonomic assignments of the trimmed sequencing reads were performed using Kraken2 against the SILVA reference database (Release 138.1, https://www.arb-silva.de/documentation/release-138/, accessed on 20 September 2022), with default parameters. The Bioconductor R package phyloseq (version 1.38.0) was employed to analyze the microbiome profile data. Sequences that were not classified as bacteria or lacked annotation at the phylum level were filtered out. Samples with less than 20,000 reads and rarely observed phylotypes were excluded from the subsequent analysis. The alpha diversity indices of the samples were calculated using the Bioconductor R package microbiome (version 1.16.0). Using Mothur (version 1.48.0), we performed calculations of beta diversity indices and conducted permutational multivariate analysis of variance (PERMANOVA) tests based on distance matrices to assess the microbiome composition differences between different phenotypes.

### 2.4. Machine Learning for Diagnosis Model

Before the machine learning model’s development, log normalization with a pseudocount of 1 was performed to reduce the difference by sequencing depth for each sample, and ANCOM-BC (version 1.4.0) was used to correct the sequencing batch effects for all samples processed in different periods.

For performing principal component analysis (PCA) and developing the diagnosis models, we utilized the Bioconductor R package mixOmics (version 6.18.1). Specifically, we employed sparse partial least squares discriminant analysis (sPLS-DA), which offers computational efficiency and generates interpretable results through graphical outputs. The feature selection and parameter optimization process followed a general procedure. To determine the optimal number of components, we assessed the performance of the PLS-DA model using 50 repeated fivefold cross-validations (5-CVs) and monitored the overall error rate trend across the CVs. Subsequently, tuning processes were conducted to simultaneously select the optimal number of components and the optimal features for each component. The final sparse PLS-DA model was defined as the one that yielded the lowest overall error rate, indicating the best predictive performance.

To assess the performance of the machine learning models, we repeated the abovementioned process 100 times by randomly splitting the samples into 70% training and 30% validation sets. Using the 100 trained sPLS-DA models, the phenotypes of the corresponding validation sets were predicted, and the average performance was measured.

## 3. Results

In our study, we performed 16S rRNA sequencing on saliva samples collected from a total of 402 individuals, which included three different disease phenotypes: Crohn’s disease (CD: *n* = 127), ulcerative colitis (UC: *n* = 175), and healthy controls (HC: *n* = 100). The baseline demographics and clinical characteristics of the patients can be found in Table 1.

During the sequencing process, a total of 54,553,630 paired-end reads were generated. After applying quality trimming, we observed an average of 121,689 paired-end reads that were retained for each sample. Taxonomy assignment and subsequent phylotyping analysis allowed us to identify 2711 phylotypes within the bacterial kingdom. These phylotypes were annotated with specific phylum information and were observed more than 10 times in our dataset.

In the subsequent analysis, we focused on 398 samples that had paired-end read counts exceeding 20,000. These samples were selected to ensure sufficient data quality and were used to investigate the microbial patterns and associations related to the different disease phenotypes.

### 3.1. Diversity Analysis

The alpha diversity analysis revealed that HC individuals had a significantly richer microbiome compared to the UC patients (*p* < 0.01) and the CD patients (*p* < 0.05), as shown in Figure 1a–c. However, there were no significant differences in alpha diversity indices between CD and UC, except for Chao1.

In all three beta diversity principal coordinates analysis (PCoA) plots of the phylotype distribution, the HC samples were stretched along PCoA2 (Figure 1d–f). While some IBD samples overlapped with HC, the others were spread far along PCoA1.

These results from the beta diversity analysis align with the taxonomic distribution depicted in Figure 2. The *Lachnospiraceae* family was the most abundant family across all three groups. Although the HC group exhibited a significantly lower abundance of the *Flavobacteriaceae* family compared to IBD, no substantial taxonomic differences were observed among the IBD subgroups at the family level. The top 10 family groups are listed based on their abundance.

### 3.2. Multi-Class Machine Learning Model Based on PLS-DA

To identify the main sources of variation, we conducted an initial analysis using principal component analysis (PCA) on the normalized log-transformed abundance values of the saliva microbiome data obtained from ANCOM-BC. The PCA results revealed the presence of two clusters; however, these clusters were not associated with the disease phenotype, consistent with the findings from the beta diversity PCoA plots. Specifically, there was no clear separation or clustering of the HC, CD, and UC samples along the first two principal components of the data (Figure 3a).

To develop a diagnostic model, we initially constructed a partial least squares discriminant analysis (PLS-DA) model capable of classifying the multi-class phenotypes. We examined the sample plot generated from this model (Figure 3b). While the plot displayed a slightly improved clustering of samples based on their phenotypes compared to the PCA plot, it remained challenging to distinguish between the three phenotypes using a single model alone.

Nevertheless, we proceeded to develop a multi-class machine learning model using the sparse PLS-DA method. The entire dataset was divided into training and test sets using a balanced split scheme, with 70% of the samples assigned to the training set (CD: *n* = 87, UC: *n* = 122, HC: *n* = 69) and 30% assigned to the test set (CD: *n* = 37, UC: *n* = 53, HC: *n* = 30). This process was repeated 100 times with different splits to ensure robustness.

In each run, we applied PLS-DA to the training set, determining the optimal number of components based on the trend of the overall error rate. Feature selection was performed through tuning processes to identify the most relevant components and phylotypes. The final sPLS-DA model for each run was defined using these optimal components and phylotypes. The performance of each model was then evaluated using the corresponding test set.

Table 2 presents the performance metrics from the 100 runs. Overall, the multi-class model demonstrated good classification of the healthy control (HC) group, but it did not perform as well for the IBD phenotypes.

### 3.3. Hierarchical Diagnosis Models Based on PLS-DA and sPLS-DA

In the previous step, it was observed that the multi-class model did not perform optimally in distinguishing all three phenotypes. To improve the accuracy of classification, we made the decision to develop two separate diagnostic models: one model for distinguishing between IBD and HC samples, and another model for classifying IBD samples into CD or UC. By employing these two models in a hierarchical manner, it becomes possible to classify an input sample into one of the three phenotypic classes.

#### 3.3.1. Development of a Prediction Model That Classifies IBD vs. HC

The binary classification dataset was generated by combining the CD and UC classes into a single IBD class, while keeping the HC class unchanged. This merging operation was performed separately for both the training set (70%: HC, *n* = 69; IBD, *n* = 209) and the test set (30%: HC, *n* = 30; IBD, *n* = 90). The entire machine learning process, including model training and evaluation, was applied to each of the 100 different splits described earlier. Appendix A displays the top 10 genera that were selected with high frequency in the final models. The performance of the models was then averaged across these splits to obtain an overall assessment of their effectiveness.

A sample plot from one of the 100 final models demonstrated a clear separation between IBD and HC samples, although there was some overlap observed (Figure 4a). To evaluate the performance of each model, we predicted the phenotypes of individuals in the respective test sets. The classification of IBD versus HC achieved a mean accuracy of 0.908 (0.808–0.967), sensitivity of 0.919 (0.8–1), specificity of 0.974 (0.633–1), precision of 0.957 (0.887–1), and AUC of 0.967 (0.916–0.995) (Table 3). The AUROC plot of the test set prediction results for one of the 100 models is shown in Figure 4b. These performance metrics indicate that the models achieved high accuracy in distinguishing between IBD and HC samples.

#### 3.3.2. Development of a Prediction Model That Classifies CD vs. UC

For the development of the model that distinguishes IBD samples into CD or UC, we kept only the CD and UC samples from the original 100 splits of the training (70%: CD, *n* = 87; UC, *n* = 122) and test (30% CD, *n* = 37; UC, *n* = 53) sets. Using this subset of data, sPLS-DA models were constructed for each split following the same procedure as described earlier. In Appendix A, the top 10 genera, which were chosen with high frequency in the final models, are presented. This entire process was repeated 100 times, and the average performance was evaluated.

A representative PLS projection plot showed a clear distinction between CD and UC individuals (Figure 5a). This indicates that the model was able to effectively differentiate between the two subtypes of IBD based on the microbiome data.

To assess the performance of the trained sPLS-DA models, the phenotypes of individuals in the corresponding test sets were predicted. The classification results from the 100 test sets showed a mean accuracy of 0.846 (0.744–0.922), sensitivity of 0.82 (0.595–0.973), specificity of 0.864 (0.679–0.962), precision of 0.812 (0.646–0.936), and AUC of 0.923 (0.829–0.978) (Table 4). Figure 5b displays the AUROC plot illustrating the prediction results of the test set from one of the 100 models. These performance metrics demonstrate that the models were capable of classifying IBD into CD and UC with a considerable level of accuracy.

#### 3.3.3. Evaluation of the Hierarchical Model

The two models were combined into a hierarchical model that classifies an input sample into three classes. In the first step, the samples are classified as either IBD or HC. Only the samples classified as IBD in the first step are further classified into CD or UC. We evaluated the performance of this model using the test sets. The results presented in Table 5 indicate that the mean accuracy, HC sensitivity, HC precision, CD sensitivity, CD precision, UC sensitivity, and UC precision were 0.803 (0.658–0.892), 0.875 (0.633–1), 0.791 (0.581–1), 0.791 (0.595–0.919), 0.792 (0.6–0.968), 0.77 (0.547–0.925), and 0.833 (0.682–0.932), respectively.

## 4. Discussion

This study revealed distinct taxonomic distributions in saliva samples between IBD patients and healthy controls, as well as the ability to differentiate between patients with Crohn’s disease (CD) and ulcerative colitis (UC). Exploring the oral microbiome as a potential diagnostic tool for IBD offers several advantages over fecal microbiome analysis. Firstly, collecting oral samples is less invasive and generally more acceptable to patients compared to collecting fecal samples. Secondly, the oral microbiome exhibits greater stability over time and is less likely to be influenced by short-term dietary changes, potentially providing a more reliable marker of long-term microbial trends. Lastly, utilizing the oral microbiome for early IBD diagnosis allows for prompt intervention before significant gut damage occurs. Hence, we opted to utilize saliva samples instead of fecal samples in our study.

We successfully identified a total of 2711 phylotypes through taxonomic assignment of 16S rRNA sequencing data from saliva samples using Kraken2 and a series of filtering processes. The number of taxa classified in our study is sufficient to assess microbial diversity and compares favorably to previous studies [23,24]. Consistent with previous research [25,26], our findings indicate that the oral microbiome of IBD patients differs from that of healthy controls in terms of both alpha and beta diversity. Regarding alpha diversity analysis, we observed a significant difference in microbial richness between the healthy controls (HCs) and patients with ulcerative colitis (UC) (*p* < 0.01), as well as between HCs and patients with Crohn’s disease (CD) (*p* < 0.05). These results suggest that the oral microbiome of HC individuals is more diverse compared to those of UC and CD patients. Furthermore, our beta diversity analysis revealed heterogeneous microbiome profiles among IBD patients. While some IBD samples exhibited similarities to HC, others showed distinct differences. This indicates that the overall composition of the oral microbiome can vary significantly in patients with IBD, with some individuals displaying a profile closer to that of HC and others displaying a distinct microbial pattern. Overall, these findings provide valuable insights into the oral microbiome characteristics associated with IBD and highlight the complexity of the microbial changes observed in this condition.

The oral cavity indeed exhibits lower microbial diversity compared to the gut. The gut microbiota is known for its remarkable diversity, with hundreds to thousands of different bacterial species. In contrast, the oral cavity faces unique challenges, including exposure to air, fluctuations in temperature and pH, and limited availability of nutrients between meals. These factors contribute to a narrower range of microbes that can thrive in the oral environment [27]. Additionally, saliva contains antimicrobial compounds such as lysozyme and lactoferrin, which can restrict microbial growth. Despite its lower taxonomic diversity, the oral microbiome plays crucial roles in maintaining overall health and can contribute to disease development. It is an integral part of the human microbiota [28]. It is important to note that although the gut microbiota generally exhibits higher diversity, the oral microbiota can vary significantly between individuals and can be influenced by factors such as oral hygiene practices, diet, and tobacco smoking. These factors can further contribute to the interindividual variability observed in the oral microbiome composition.

Machine learning (ML) is a subfield of artificial intelligence (AI) that empowers computers to learn from data and make predictions or take actions without explicit programming. ML has found applications in various domains, including healthcare. In the context of IBD diagnosis, ML techniques can be utilized with diverse data sources, such as endoscopic images, histopathology slides, blood tests, fecal biomarkers, and electronic health records [29]. The gut microbiome, in particular, holds great promise as a valuable data source for ML-based approaches in IBD. It represents the complex interplay between the host and the environment, offering valuable insights into the pathogenesis and prognosis of the disease. By leveraging ML algorithms, researchers and clinicians can analyze and interpret the vast amounts of microbial data to identify patterns, biomarkers, and predictive models that aid in the diagnosis, treatment, and prognosis of IBD.

ML can be applied to analyze the microbiome data for IBD diagnosis. ML can be used to (1) classify samples into different microbiome groups based on their composition or function, (2) predict continuous outcomes based on the microbiome features, (3) group samples into clusters based on their microbiome similarity, (4) reduce the complexity and noise of high-dimensional microbiome data, and (5) discover associations between the microbiome and other variables of interest. Several studies [17,19,30,31,32] have evaluated the performance of ML models for microbiome analysis in IBD diagnosis using different datasets, methods, and metrics. Overall, ML models have demonstrated good accuracy, sensitivity, specificity, and area under the curve (AUC) for IBD diagnosis.

In our study, we employed sPLS-DA, which is a computationally efficient method that yields interpretable results through graphical outputs. sPLS-DA is particularly valuable in high-dimensional datasets, where numerous variables may be irrelevant or redundant. When applied to microbiome data, sPLS-DA can help identify a subset of microbial species that exhibit the highest predictive power for a specific outcome, such as disease status (e.g., IBD vs. healthy control). The key advantage of utilizing sPLS-DA in this context is its ability to handle high-dimensional data, effectively manage correlations, and identify the most significant microbial species for predicting the outcome.

While machine learning (ML) holds significant potential for improving the diagnosis of IBD patients based on their microbiome data, there are several challenges and limitations that must be addressed before widespread adoption of ML in clinical practice. Firstly, ML models rely on large and diverse datasets that accurately represent the target population, with well-annotated, standardized, and validated data. Without such robust datasets, the performance and generalizability of ML models can be compromised. Secondly, ML models often exhibit complexity and opaqueness, making it challenging to comprehend how they arrive at decisions and which features they prioritize. This lack of interpretability can hinder trust and acceptance from clinicians and patients, as well as impeding error identification and bias mitigation. Thirdly, ML models necessitate testing and validation on independent and external datasets that encompass diverse settings, populations, and scenarios. This ensures the robustness and reliability of the models in real-world applications. Lastly, the integration of ML models into existing clinical workflows and systems, such as electronic health records or mobile applications, is crucial. Furthermore, evaluating the clinical impact, cost-effectiveness, and user satisfaction of ML models is necessary to gauge their overall utility and practicality. Addressing these challenges and limitations will be pivotal in harnessing the full potential of ML for enhancing the diagnosis and management of IBD patients based on their microbiome data.

Hence, future research on ML for microbiome analysis in IBD diagnosis should focus on the following: (1) Developing and sharing high-quality datasets for microbiome analysis in IBD diagnosis that are large, diverse, standardized, and annotated. (2) Applying explainable AI techniques to enhance the transparency and interpretability of ML models for microbiome analysis in IBD diagnosis. (3) Conducting rigorous external validation and clinical trials to assess the performance and utility of ML models for microbiome analysis in IBD diagnosis. (4) Collaborating with multidisciplinary teams to design and implement user-friendly and clinically relevant ML solutions for microbiome analysis in IBD diagnosis.

As we found that sPLS-DA models with differently divided training sets were able to distinguish IBD vs. HC with high performance (average accuracy = 0.908, sensitivity = 0.919, precision = 0.957, AUC = 0.966) and distinguish CD vs. UC with average accuracy 0.846, sensitivity 0.82, precision 0.812, and AUC = 923 in the prediction of the corresponding test sets, our study employed machine learning techniques to identify potential oral microbial markers to distinguish not only IBD and HC, but also CD and UC. The distinctive microbial patterns associated with each disease can provide valuable insights for their differential diagnosis.

Machine learning has revolutionized how we analyze complex biological data, offering an unparalleled ability to detect patterns and correlations within highly multidimensional datasets [33]. In this study, we employed machine learning to discern between CD and UC, focusing on the oral microbiome. This strategy yielded promising results, suggesting potential for the application of such techniques in differential diagnosis. The success of this study is primarily premised on the growing evidence that the human microbiome—and particularly the oral microbiome—has implications for IBD.

However, it is important to acknowledge the limitations of our research. Firstly, we were not able to identify a specific strain that was consistently associated with IBD patients. This highlights the complexity of the oral microbiome and its interactions with the gut microbiome, which are still not fully understood. While our study identified potential microbial markers, further investigations are needed to elucidate their functional implications and the underlying mechanisms linking these oral microbial markers to the development of CD and UC. Additionally, although machine learning provided valuable insights in our study, its widespread adoption in routine clinical practice is hindered by the requirement for significant computational resources and expertise. Overcoming these challenges and refining these tools to enhance accessibility and ease of use will be crucial for their practical implementation [34,35]. Finally, although the purpose of this study was to assess the predictive performance of the saliva microbiome alone, it would be highly reasonable to improve the model’s performance by incorporating additional types of data, such as clinical data. Therefore, we intend to pursue this approach in future investigations.

In summary, while our research presents early promise for the use of oral microbial markers in the diagnosis and differentiation of IBD, there is a need for further research to address the limitations and fully explore the functional significance of these markers. Furthermore, efforts should be made to optimize machine learning techniques for clinical application in order to realize their potential in improving patient care.

## 5. Conclusions

Our study offers preliminary but promising evidence regarding the potential usefulness of oral microbial markers in diagnosing IBD compared to healthy controls, as well as in differentiating between CD and UC. The application of machine learning models has facilitated these advancements. The development of noninvasive diagnostic methods, based on these findings, could greatly enhance our ability to diagnose, distinguish, and effectively manage IBD. This, in turn, could pave the way for personalized medicine approaches in the field of IBD.

## Figures and Tables

**Figure 1 microorganisms-11-01665-f001:**
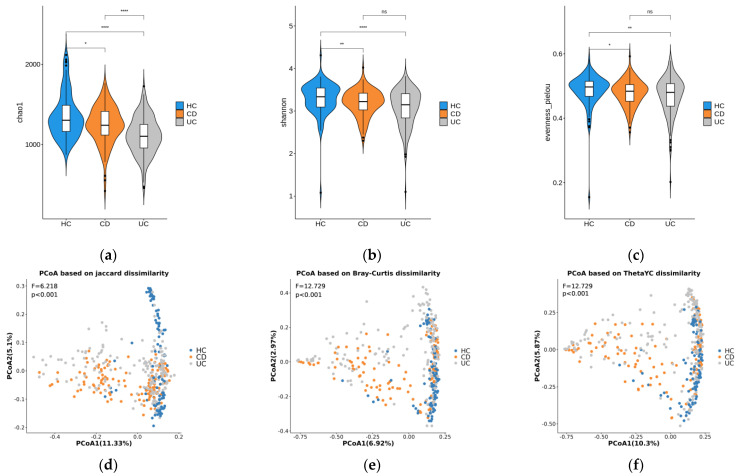
Diversity plots. Alpha diversity indices (Chao1 (**a**), Shannon (**b**), and Pielou evenness (**c**)) of the microbial communities in the three disease phenotype groups are represented as violin plots. The Wilcoxon test was used to measure the significance of the alpha diversity differences between disease phenotypes. *, **, ****, and ns correspond to *p* < 0.05, *p* < 0.01, *p* < 0.0001, and non-significance, respectively. PCoA plots were based on three beta diversity indices: Jaccard dissimilarity (**d**), Bray−Curtis dissimilarity (**e**), and the Yue and Clayton measure of dissimilarity (**f**). F- and *p*-values were calculated by a PERMANOVA test with 1000 permutations.

**Figure 2 microorganisms-11-01665-f002:**
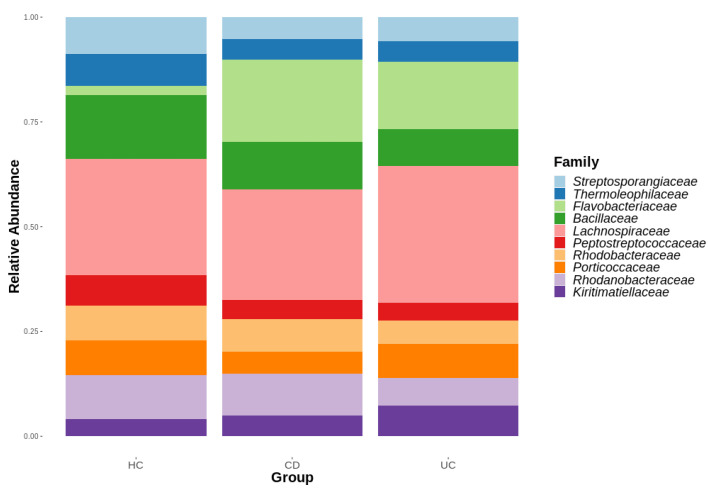
Taxonomic composition bar plot of saliva samples from HC, CD, and UC groups at the family level.

**Figure 3 microorganisms-11-01665-f003:**
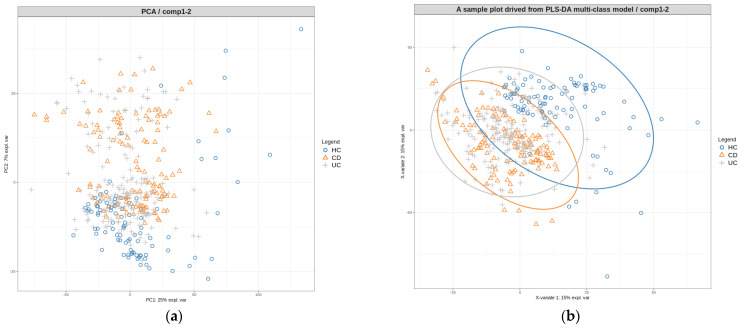
Principal component analysis (**a**) and PLS−DA (**b**) plots of saliva microbiome data. Normalized log-transformed abundance values from ANCOM−BC were used.

**Figure 4 microorganisms-11-01665-f004:**
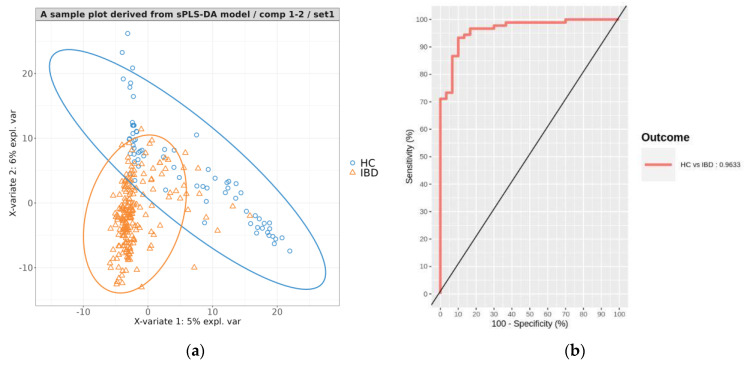
(**a**) A representative PLS projection onto the subspace spanned by the two components of the sPLS−DA model developed for classifying IBD vs. HC. The ellipses for each class represent the 95% confidence levels of discrimination. (**b**) Area under the receiver operating characteristic curve (AUROC) for the predictions on the validation set of the representative model.

**Figure 5 microorganisms-11-01665-f005:**
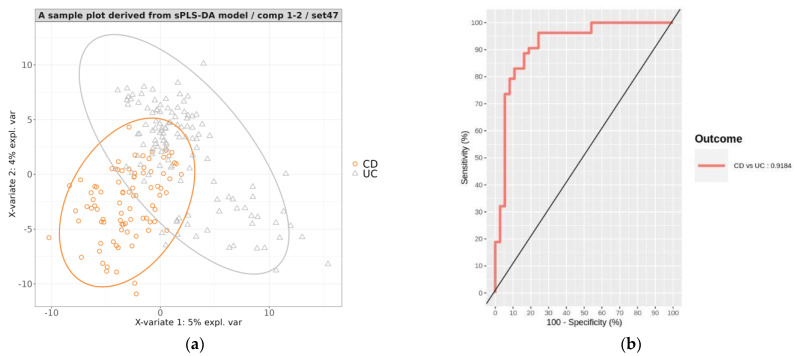
(**a**) A representative PLS projection onto the subspace spanned by the two components of the sPLS−DA model developed for classifying CD and UC. The ellipses for each class represent the 95% confidence levels of discrimination. (**b**) Area under the receiver operating characteristic curve (AUROC) for the predictions on the test set of the representative model.

**Table 1 microorganisms-11-01665-t001:** Baseline demographic and clinical characteristics of participants.

	CD (*n* = 127)	UC (*n* = 175)	HC (*n* = 100)
Age (year), mean ± SD	37.6 ± 11.6	39.4 ± 15.6	37.4 ± 14.5
Male, n (%)	97 (76.4)	124 (70.9)	50 (50)
BMI (kg/m^2^), mean ± SD	20.8 ± 4.1	23 ± 3.1	
Smoking status, n (%)			
Current	20 (15.7)	23 (13.1)	
Former	5 (3.9)	37 (21.1)	
Never	101 (79.5)	112 (64)	
Unknown	1 (0.8)	3 (1.7)	
Disease location, n (%)	Ileum, 28 (22)	Proctitis, 69 (39.4)	
	Colon, 25 (19.7)	Distal, 57 (32.6)	
	Ileocolon, 71 (55.9)	Extensive, 45 (25.7)	
	Ileocolon + upper GI, 1 (0.8)		
	Unknown, 2 (1.6)	Unknown, 4 (2.3)	

SD, standard deviation; BMI, body mass index.

**Table 2 microorganisms-11-01665-t002:** Evaluation metrics from prediction using multi-class models.

	Accuray	CD Sens.	CD Prec.	UC Sens.	UC Prec.	HC Sens.	HC Prec.	AUC
Min.	0.525	0.405	0.442	0.377	0.574	0.633	0.476	0.650
1st Qu.	0.617	0.595	0.565	0.491	0.665	0.8	0.619	0.759
Median	0.658	0.676	0.624	0.528	0.705	0.867	0.659	0.801
Mean	0.653	0.666	0.615	0.532	0.7	0.85	0.652	0.819
3rd Qu.	0.683	0.73	0.659	0.566	0.738	0.9	0.693	0.898
Max.	0.758	0.919	0.808	0.755	0.852	1	0.839	0.968

Sens, sensitivity; Prec, precision. Evaluation metrics were calculated from each of 100 models.

**Table 3 microorganisms-11-01665-t003:** Evaluation metrics from prediction using IBD vs. HC models.

	Accuracy	Sensitivity	Specificity	Precision	AUC
Min.	0.808	0.8	0.633	0.887	0.916
1st Qu.	0.892	0.9	0.833	0.943	0.954
Median	0.908	0.917	0.867	0.955	0.967
Mean	0.908	0.919	0.974	0.957	0.966
3rd Qu.	0.925	0.944	0.933	0.977	0.979
Max.	0.967	1	1	1	0.995

Evaluation metrics were calculated from each of 100 models.

**Table 4 microorganisms-11-01665-t004:** Evaluation metrics from prediction using CD vs. UC models.

	Accuracy	Sensitivity	Specificity	Precision	AUC
Min.	0.744	0.595	0.679	0.646	0.829
1st Qu.	0.822	0.784	0.83	0.767	0.899
Median	0.844	0.824	0.87	0.816	0.928
Mean	0.846	0.82	0.864	0.812	0.923
3rd Qu.	0.878	0.872	0.906	0.857	0.949
Max.	0.922	0.973	0.962	0.936	0.978

Evaluation metrics were calculated from each of 100 models.

**Table 5 microorganisms-11-01665-t005:** Evaluation metrics from prediction using hierarchical models.

	Accuray	CD Sens.	CD Prec.	UC Sens.	UC Prec.	HC Sens.	HC Prec.
Min.	0.658	0.595	0.6	0.547	0.682	0.633	0.581
1st Qu.	0.783	0.73	0.738	0.717	0.799	0.833	0.741
Median	0.8	0.797	0.795	0.764	0.837	0.867	0.784
Mean	0.803	0.791	0.792	0.77	0.833	0.875	0.791
3rd Qu.	0.825	0.865	0.844	0.83	0.867	0.933	0.839
Max.	0.892	0.919	0.968	0.925	0.932	1	1

Sens, sensitivity; Prec, precision. Evaluation metrics were calculated from each of 100 models.

## Data Availability

The raw 16S rRNA amplicon sequences generated in this study have been deposited in the Korean Nucleotide Archive (KoNA) operated by the Korea Bioinformation Center (KOBIC) under accession code PRJKA230590.

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
