# Peer review of "Potential Oral Microbial Markers for Differential Diagnosis of Crohn’s Disease and Ulcerative Colitis Using Machine Learning Models"

_microorganisms, 2023, doi:10.3390/microorganisms11071665_

Round 1

Reviewer 1 Report

The study by Kang et al. concerns the use of ML and oral microbiome analysis for the differential diagnosis of Crohn’s disease and ulcerative colitis.

The study is straightforward, and the manuscript is well-written.

I have two major suggestions/questions for the authors’ consideration:

1.    It would be important if you could provide the identity of at least some of the taxons that allowed the discrimination between the different groups.

 2.    Given the time required from sample to microbiome analysis additional routine clinical samples may be analyzed.  For example, results from blood tests or other can also be obtained. Why did you not consider a data fusion approach in which you could merge multiple data sources (including your microbiome data) to test if you would derive to better models for the differential diagnosis of the IBD than the ones you currently report?

Minor comments:

Line 58-59: “Dysbiosis not only affects the gut but can also occur in the oral cavity, potentially impacting the function of the gut barrier and immune response, which are key factors in the development of chronic inflammation, a hallmark of IBD 60 [10,11].” The meaning of this sentence is unclear. In the way it is currently written, the sentence seems to imply that dysbiosis in the oral cavity may have an impact on the gut barrier. This is a bit odd.

Line 253: Figure 4 (there is no 4a)

Author Response

Response to Reviewer 1 Comments

Point 1: It would be important if you could provide the identity of at least some of the taxons that allowed the discrimination between the different groups.

Response 1: We agree with your opinion and have added tables in the supplementary materials that includes some of taxon allowing differentiation between different groups.

Point 2: Given the time required from sample to microbiome analysis additional routine clinical samples may be analyzed.  For example, results from blood tests or other can also be obtained. Why did you not consider a data fusion approach in which you could merge multiple data sources (including your microbiome data) to test if you would derive to better models for the differential diagnosis of the IBD than the ones you currently report?

Response 2: It is highly reasonable to create a better-performing model by incorporating other types of data, and we agree with your opinion and plan to pursue it in the future. However, the purpose of this paper is to evaluate the performance of the Saliva microbiome alone. We have mentioned this matter in the discussion section.

Minor comments 1: Line 58-59: “Dysbiosis not only affects the gut but can also occur in the oral cavity, potentially impacting the function of the gut barrier and immune response, which are key factors in the development of chronic inflammation, a hallmark of IBD 60 [10,11].” The meaning of this sentence is unclear. In the way it is currently written, the sentence seems to imply that dysbiosis in the oral cavity may have an impact on the gut barrier. This is a bit odd.

Minor comments Response 1: We have made modifications to the manuscript regarding that sentence.

Minor comments 2: Line 253: Figure 4 (there is no 4a)

Minor comments Response 2: During the handling of the reviewer's comments, an additional plot were added. The corresponding figure number was changed to 5a.

Reviewer 2 Report

Kang et al. used a machine learning model to analyze oral microbiomes in patients with Crohn's disease and ulcerative colitis, which provided insight into potential diagnostic markers for IBD. The manuscript is well-written and contains some important findings. However, there are a few points that need to be clarified in the manuscript. Here are some comments on this paper and questions that I would like to discuss with the authors:

1.        Line 33 (sPLS-DA) needs to be replaced with (Sparse Partial Least Squares Discriminant Analysis, sPLS-DA) in the abstract.

2.        Could the authors provide more detail on how the saliva samples were collected? I believe that the sample collection method is a critical issue for oral microbiome study.

3.        Could authors upload the raw sequencing data to NCBI? Because public sharing of sequencing data is an important part of microbiome research.

4.        Line 146-150, the font size is not the same as others.

5.        The BMI information of the HC group is missing in Table 1.

6.        In section 3.1, could authors include PERMANOVA analysis in the PCoA plots?

7.        Before section 3.2, could the authors provide some bar plots to show the structure of the microbiome in the three groups? It will help us to better understand the composition of the oral microbiome in patients with IBD.

8.        Although the authors mentioned oral microbial markers several times in the manuscript, they did not specify what the markers were. As far as I know, the sPLS-DA model could depict the weight of each variable (oral microbiome), which allows for identifying the oral microbial markers.

9.        Would it be better to use the AUC plot instead of the table to present the sPLS-DA results?

10.     There are a lot of discussions (lines 334-341) about machine learning in the discussion section. Since machine learning is not the main focus of this paper, it is desirable to reduce some discussions about machine learning.

11.     There are some discrepancies in the format of references.

Author Response

Response to Reviewer 2 Comments

Point 1: Line 33 (sPLS-DA) needs to be replaced with (Sparse Partial Least Squares Discriminant Analysis, sPLS-DA) in the abstract.

Response 1: We have made the revisions to the manuscript according to your comments.

Point 2: Could the authors provide more detail on how the saliva samples were collected? I believe that the sample collection method is a critical issue for oral microbiome study.

Response 2: We have addressed the comment by adding additional details on how the saliva samples were collected in the manuscript. We recognize that the sample collection method is a critical aspect for oral microbiome studies.

Point 3: Could authors upload the raw sequencing data to NCBI? Because public sharing of sequencing data is an important part of microbiome research.

Response 3: We agree with your suggestion, and for the public sharing of sequencing data, we have uploaded the raw sequencing data used in our study to the Korean Nucleotide Archive (KoNA), operated by the Korea Bioinformation Center (KOBIC). We have also included this information in the "Data availability" section of the manuscript.

Point 4: Line 146-150, the font size is not the same as others.

Response 4: We have made the font size of that section of the manuscript consistent with the others.

Point 5: The BMI information of the HC group is missing in Table 1.

Response 5: We did not measure the height and weight of the healthy controls, so we were unable to determine their BMI.

Point 6: In section 3.1, could authors include PERMANOVA analysis in the PCoA plots?

Response 6: Taking your feedback into consideration, we have incorporated the results of the beta-diversity index-based PERMANOVA test into the PCoA plots presented in Section 3.1.

Point 7: Before section 3.2, could the authors provide some bar plots to show the structure of the microbiome in the three groups? It will help us to better understand the composition of the oral microbiome in patients with IBD.

Response 7: Based on your feedback, we have included additional plots in the manuscript

Point 8: Although the authors mentioned oral microbial markers several times in the manuscript, they did not specify what the markers were. As far as I know, the sPLS-DA model could depict the weight of each variable (oral microbiome), which allows for identifying the oral microbial markers.

Response 8: We agree with your opinion and have added tables in the supplementary materials that includes some of taxon allowing differentiation between different groups

Point 9: Would it be better to use the AUC plot instead of the table to present the sPLS-DA results?

Response 9: We added a summary of the AUC values for the test set predictions of the 100 models to the table(2, 3, 4), and included the AUC plot of the representative model in each corresponding section. In the case of hierarchical models, it was not possible to calculate the AUC because the number of classification occurrences varies for each sample.

Point 10: There are a lot of discussions (lines 334-341) about machine learning in the discussion section. Since machine learning is not the main focus of this paper, it is desirable to reduce some discussions about machine learning.

Response 10: We agree with your opinion, and we have condensed the discussion on machine learning in the discussion section.

Point 11: There are some discrepancies in the format of references.

Response 11: We have corrected the inconsistencies in the reference format in the manuscript.

Round 2

Reviewer 1 Report

The authors have addressed the reviewers' comments